# Alternating L4 loop architecture of the bacterial polysaccharide co-polymerase WzzE

Benjamin Wiseman [1✉], Göran Widmalm [2] & Martin Högbom [1✉]

Lipopolysaccharides such as the enterobacterial common antigen are important components of the enterobacterial cell envelope that act as a protective barrier against the environment and are often polymerized by the inner membrane bound Wzy-dependent pathway. By employing cryo-electron microscopy we show that WzzE, the co-polymerase component of this pathway that is responsible for the length modulation of the enterobacterial common antigen, is octameric with alternating up-down conformations of its L4 loops. The alternating up-down nature of these essential loops, located at the top of the periplasmic bell, are modulated by clashing helical faces between adjacent protomers that flank the L4 loops around the octameric periplasmic bell. This alternating arrangement and a highly negatively charged binding face create a dynamic environment in which the polysaccharide chain is extended, and suggest a ratchet-type mechanism for polysaccharide elongation.

[1] Department of Biochemistry and Biophysics, Stockholm University, Stockholm, Sweden. [2] Department of Organic Chemistry, Stockholm University, Stockholm, Sweden. ✉email: benjamin.wiseman@dbb.su.se; hogbom@dbb.su.se

Serotype specific lipopolysaccharides (LPS) and the enterobacterial common antigen (ECA) are important components of the outer membrane of Gram-negative bacteria that play a vital role in many biological processes such as maintaining outer membrane homeostasis, resistance to bile salts[1], biofilm formation, pathogenesis, immune evasion and act as a protective barrier against the environment and toxic molecules such as antibiotics[2]. One LPS biosynthesis pathway is known as the Wzy-dependent pathway where O-antigen (Oag) and ECA polymers are synthesized directly at the inner membrane of Gram-negative bacteria[3–5], where WzyB/WzzB is responsible for Oag synthesis and WzyE/WzzE is responsible for ECA synthesis as well as being required for synthesis of the cyclic forms of ECA[6].

As a key component of the Wzy-dependent biosynthesis pathway, WzzE is a polysaccharide co-polymerase responsible for modulating the length distribution of the synthesized polysaccharide chain by interacting with the membrane bound polymerase WzyE as it polymerizes the growing ECA polysaccharide[4]. Polysaccharide co-polymerases are known to homo-oligomerise into a unique structure resembling a box jellyfish[7] containing a large bell-shaped periplasmic domain[8–12] with the transmembrane domains encircling a large chamber[7]. At the top of the bell-shaped periplasmic domain, located approximately 80 Å away from the membrane interface, is the highly flexible L4 loop, that despite being located far from the membrane interface is vitally important in the production of long polysaccharide chains[10,13,14]. It has been suggested that the flexibility of the L4 loop could mediate conformational changes associated with function[10,14]. Due to the presumed flexibility of this loop, it is not surprising that the majority of the available structures contain either a distorted or incomplete L4 loop, or is missing the loop all-together in the deposited models[8].

With that in mind, using single-particle cryo-electron microscopy we determined the structure and show that the polysaccharide co-polymerase WzzE is octameric with an alternating up-down pattern of its L4 loops that are located at the top of the periplasmic bell approximately 100 Å from the membrane interface. Our analysis reveals that this alternating up-down pattern is a robust, conserved feature of WzzE that is governed by global structural forces. In addition, the alternating nature of the L4 loops create a highly negatively charged potential sugar binding face and using 3D variable analysis we are able to visualize directly in the single-particle data a back-and-forth movement of the L4 loops and flanking helices. Overall, we show that the WzzE complex creates a highly dynamic, negatively charged environment where the polysaccharide chain could be elongated in a potential ratchet-like mechanism.

## Results and discussion

**WzzE is a C4 symmetric octamer with alternating arrangements of its L4 loops.** We purified (Supplementary Fig. 1) and determined the structure of the fully intact polysaccharide co-polymerase WzzE from *E. coli* by cryo-electron microscopy to an overall resolution of 3.2 Å. Consistent with previously determined structures of this family, an octameric bell-shaped complex was identified by reference-free ab-initio 3D classification and is clearly visible in the un-symmetrized (C1) map (Fig. 1a and Supplementary Fig. 2). An overall resolution of 3.4 Å was obtained without the application of symmetry and, although clearly octameric in nature, displayed a loss of octameric symmetry in the region of the L4 loops at the top of the periplasmic bell (Fig. 1a and Supplementary Fig. 2). In this region the L4 loops are seen in two conformations, one up and one down with, remarkably, the subunits clearly alternating between the up and

down conformations around the octameric complex. Although the application of C8 symmetry did marginally improve the overall resolution of the map to 3.1 Å, the alternating nature of the L4 region was averaged out (Fig. 1a and Supplementary Fig. 3). Thus, in order to resolve this alternating arrangement of loops, C4 symmetry was applied by using two adjacent subunits encompassing the two L4 conformations as the asymmetric unit resulting in a 3.2 Å map. The high quality of the C4 symmetrized map allowed the complete amino acid sequence of the periplasmic domain to be built (Fig. 1b and Table 1), including the dynamic L4 loops in alternating conformations.

Overall, the periplasmic domain is very similar to the previously described structures[7,8,10]: a bell-shaped domain ~100 Å in height above the membrane interface that contains a hollow interior. The bell-shaped complex results from a side-by-side packing of protomers and the interactions observed here are also similar to what have been described elsewhere[7,8,10]. This structure, however, contains alternating L4 loops located at the top of the periplasmic bell (residues 238–256). Structural alignments of two adjacent WzzE subunits to a single monomer of WzzB[7] (Fig. 2a, b) reveal highly similar structures at the single subunit level despite low identity of the amino acid sequence[4] (Supplementary Fig. 4) with the position of the L4 loop being the main difference. The downward facing L4 loop of WzzE points into the center of the periplasmic bell and resembles the position seen in WzzB, although slightly more compact. The upward facing L4 loop on the other hand, is flipped upwards and is at significantly lower resolution (Fig. 2 and Supplementary Fig. 3). As a testament to the flexibility of this region of the complex, the top of α6 of WzzB can be seen significantly bent compared to the upward and downward states of WzzE by 26° and 19°, respectively towards the center of the complex (Fig. 2b). Within WzzE, this alternating arrangement of the L4 loops results in a subtle rearrangement of its two flanking helices (α6 and α7): when in the upward facing position α6 helix is tilted approximated 9° outwards away from α7, and is slightly elongated suggesting that a slight unraveling occurs when L4 is in the downward position (Fig. 2c); and when the L4 loop is in the downward position α7 tilts slightly inwards towards the center of the periplasmic bell (Fig. 2c). Thus, as a consequence of the up-down nature of the L4 loops the full octameric structure of WzzE contains a 13 Å opening at the top of its periplasmic bell nearly double the distance compared to that of WzzB containing all downward L4 loops at 7.5 Å at their narrowest points (Fig. 3a, b).

**The alternating arrangement of the L4 loops is a conserved, robust feature of WzzE.** Despite extensive 2D and 3D classifications of the WzzE single-particle data no other conformations other than an alternating L4 could be found; i.e., no WzzE complexes containing adjacent upward or adjacent downward facing L4 loops. To better understand why L4 has to alternate between adjacent WzzE protomers we generated artificial all-up and all-down verisons of WzzE by applying C8 symmetry to a single protomer containing an upward facing L4 and a single protomer containing a downward facing L4 (Fig. 3). The naturally occurring alternating WzzE has a clash profile remarkably similar to that of the WzzB (Fig. 3a, b) with only two clashes (van der Waals distance ratio of less than 0.89) between adjacent protomers detected. The all-up and all-down versions of WzzE on the other hand, produced very different profiles. The all-up version contains no such clashes between adjacent protomers while the all-down version produced significant clashes between α6 and α7 of adjacent protomers that are absent in the alternating WzzE version. Although the all-down version of WzzE is strikingly similar to WzzB when viewed from the top of the periplasmic bell

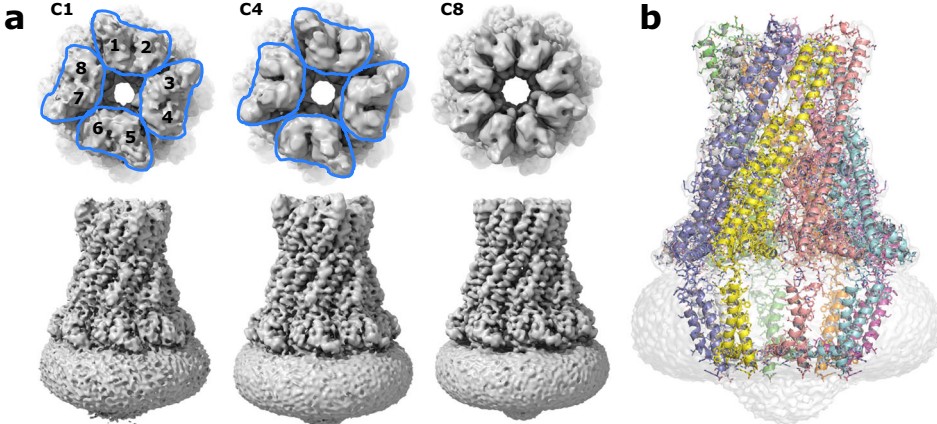

**Fig. 1 Overall structure of *E. coli* WzzE. a** Final reconstructed volumes of the full-length WzzE refined without and with the application of C4 and C8 symmetry. Top: view looking down from the top of the periplasmic domain. The application of C8 symmetry averages out the density for the alternating conformations of the L4 loops. Blue outline represents the four asymmetric units. **b** Cartoon representation of the octameric complex overlaid with the C4 symmetrized WzzE map (white).

**Table 1 EcWzzE cryo-EM data collection, refinement, and validation statistics.**

| | | EcWzzE | | EcWzzE-R267A | | EcWzzE-R267E | |
|---|---|---|---|---|---|---|---|
| | EMD-16072 | EMD-16071 PDB 8BHW | EMD-16073 | EMD-17388 | EMD-17387 PDB 8P3O | EMD-17390 | EMD-17389 PDB 8P3P |
| **Data collection and processing** | | | | | | | |
| Magnification | 130,000× | 130,000× | 130,000x | 105,000x | 105,000x | 105,000x | 105,000x |
| Voltage (kV) | 300 | 300 | 300 | 300 | 300 | 300 | 300 |
| Electron exposure (e⁻/Å²) | 52 | 52 | 52 | 50 | 50 | 50 | 50 |
| Defocus range (μm) | 1.4 to 3.2 | 1.4 to 3.2 | 1.4 to 3.2 | 0.8 to 1.9 | 0.8 to 1.9 | 0.8 to 1.9 | 0.8 to 1.9 |
| Pixel size (Å) | 1.06 | 1.06 | 1.06 | 0.868 | 0.868 | 0.828 | 0.828 |
| Extraction box size (pixels) | 400 | 400 | 400 | 470 | 470 | 500 | 500 |
| Symmetry imposed | C1 | C4 | C8 | C1 | C4 | C1 | C4 |
| Initial particle images (no.) | 578253 | 578253 | 578253 | 1198596 | 1198596 | 2349935 | 2349935 |
| Final particle images (no.) | 197015 | 197015 | 197015 | 360569 | 360569 | 308817 | 308817 |
| Map resolution (Å) | 3.5 | 3.2 | 3.1 | 3.1 | 2.9 | 2.7 | 2.5 |
| FSC threshold | 0.143 | 0.143 | 0.143 | 0.143 | 0.143 | 0.143 | 0.143 |
| Map resolution range (Å) | 9.0 to 2.3 | 6.9 to 2.3 | 6.9 to 2.3 | 8.6 to 2.1 | 6.7 to 1.7 | 6.8 to 2.1 | 6.0 to 1.5 |
| **Refinement** | | | | | | | |
| Initial model used (PDB code) | – | N/A | – | – | 8BHW | – | 8BHW |
| Model resolution (Å) | – | 3.3 | – | – | 3.0 | – | 2.7 |
| FSC threshold | – | 0.5 | – | – | 0.5 | – | 0.5 |
| Model resolution range (Å) | – | 6.0 to 2.3 | – | – | 5.5 to 1.7 | – | 5.5 to 1.5 |
| Map sharpening *B* factor (Å²) | – | 129 | – | – | 138 | – | 89.0 |
| **Model composition** | | | | | | | |
| Non-hydrogen atoms (no.) | – | 21248 | – | – | 21192 | – | 21224 |
| Protein residues (no.) | – | 2656 | – | – | 2656 | – | 2656 |
| Ligands (no.) | – | 0 | – | – | 0 | – | 0 |
| **B factors (Å²)** | | | | | | | |
| Protein | – | 190 | – | – | 165 | – | 125 |
| Ligand | – | – | – | – | – | – | – |
| **R.m.s. deviations** | | | | | | | |
| Bond lengths (Å) | – | 0.003 | – | – | 0.004 | – | 0.003 |
| Bond angles (°) | – | 0.561 | – | – | 1.031 | – | 0.626 |
| **Validation** | | | | | | | |
| MolProbity score | – | 1.55 | – | – | 1.65 | – | 1.61 |
| Clashscore | – | 6.49 | – | – | 7.03 | – | 5.98 |
| Poor rotamers (%) | – | 0.18 | – | – | 0.41 | – | 0.86 |
| **Ramachandran plot** | | | | | | | |
| Favored (%) | – | 96.82 | – | – | 96.14 | – | 95.91 |
| Allowed (%) | – | 3.18 | – | – | 3.86 | – | 4.09 |
| Disallowed (%) | – | 0.00 | – | – | 0.00 | – | 0.00 |

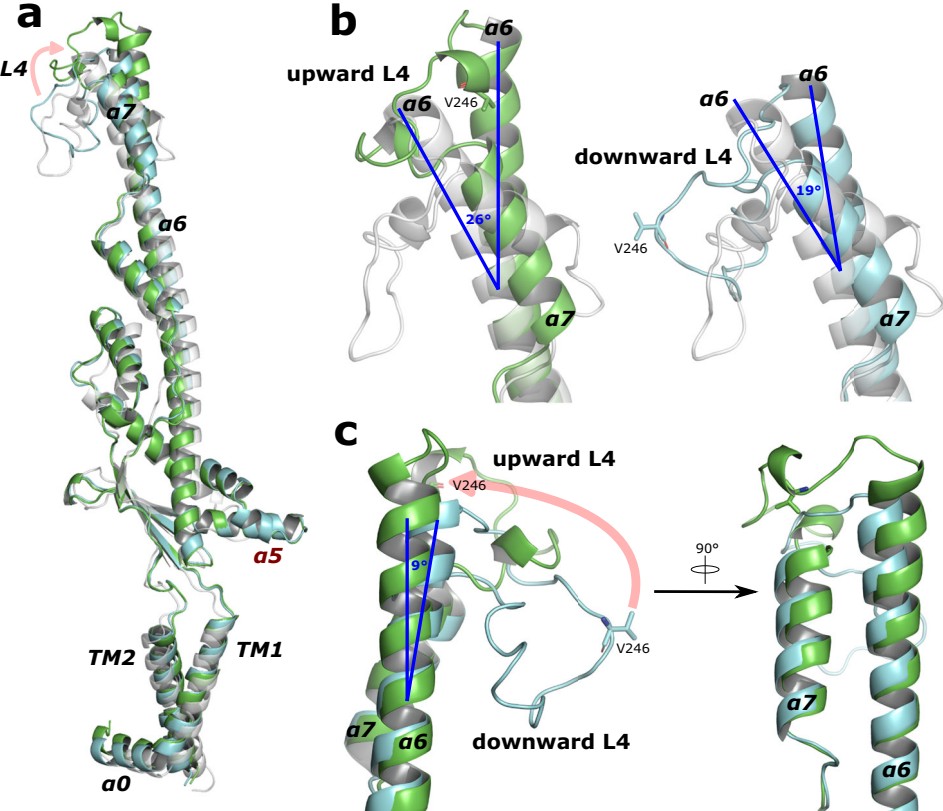

**Fig. 2 Dynamics of the L4 loop in WzzE. a** Structural alignment of adjacent WzzE protomers containing an up (green) and down (cyan) L4 loop to a protomer of WzzB (PDB id 6RBG, faint white) with labels of the α-helices and loops discussed in the text. In red, α5 is missing in WzzB. **b** Zoom of the L4 region showing its dynamics. Left: in the upwards conformation and right: in the downwards conformation aligned to the homologous region of WzzB. **c** Overlay of the two L4 loop conformation of WzzE.

with both having a diameter of 55 Å, the clashes between α6 and α7 of adjacent protomers are missing in WzzB. This can be explained by a shortened α7 helix compared to WzzE (Fig. 3c, d and Supplementary Fig. 4). The shortening of α7 and thus the removal of rigid secondary structure elements may explain why WzzB can adopt down-down L4 arrangements whereas WzzE cannot. On the other hand, why up-up versions of WzzE are not seen is not immediately evident. Possibly protomer-protomer stabilizing interactions are missing in this conformation resulting in an unstable complex or L4 loop arrangement. Thus, it is possible that the alternating arrangement strikes the perfect balance of stability to allow for a dynamic complex.

A ConSurf[15,16] analysis of WzzE revealed a similar pattern of conserved residues as seen in WzzB[7], with among others, patches of conserved residues flanking the L4 loop within α7 and the top of α6 (Fig. 4a), with the L4 loop itself containing mainly variable residues similar to WzzB. Conserved patches of residues along α7 further cement the importance of this region of the complex and suggest the elongated α7 is a distinguishing feature of WzzE and likely the driving force behind the alternating nature of the L4 loops. The significance of these results is further supported by a previous mutational study[13] in which strings of five amino acids were inserted into this region of WzzB, resulting in either the total loss of polysaccharide chain length regulation or the detection of only short polysaccharide chains.

The ConSurf analysis also revealed an arginine within α7 (R267) that interacts with the downward facing L4 (Fig. 4b) to be highly conserved (Fig. 4a and Supplementary Fig. 5). To investigate how robust the alternating conformation of the L4 loops is, we replaced this residue with an alanine and a glutamic

acid in two constructs. Although the change of this amino acid is clearly visible in the cryo-EM density map, replacement had little effect on the overall octameric structure of WzzE or the alternating arrangement of the L4 loops (Fig. 4c, d and Supplementary Fig. 6). The R267A variant did, however, have a small, but noticeable 2 Å shift upwards of the entire L4 region suggesting that this conserved residue could somehow help stabilize this otherwise dynamic region of the complex. Although there are only small differences between the native WzzE and both R267 variants, these variants do underscore the nature of the alternating loops in WzzE, as both structures of the mutants were solved independently using reference-free ab-initio starting volumes and clearly display the same pattern of alternating L4 loops in their unsymmetrized C1 maps as the native WzzE. The results of these variant proteins thus suggest that the alternating conformation of the L4 loops is a robust feature governed by global structural properties while the conservation of R267 may be related to substrate binding, as mutations to this region in WzzB resulted in the detection of only short length polysaccharide chains[13].

**The alternating L4 loops are dynamic and create a negatively charged potential sugar binding face along the interior of WzzE.** To confirm the dynamic character and alternating loop nature of this region in WzzE, a 3D variable analysis (3DVA)[17] was performed on an unsymmetrized map focused on the top of periplasmic bell (Supplementary Movie 1). Despite the small size of this region of the complex, which is likely hampering this type of analysis, clear movement of the L4 loop and flanking α6 and α7

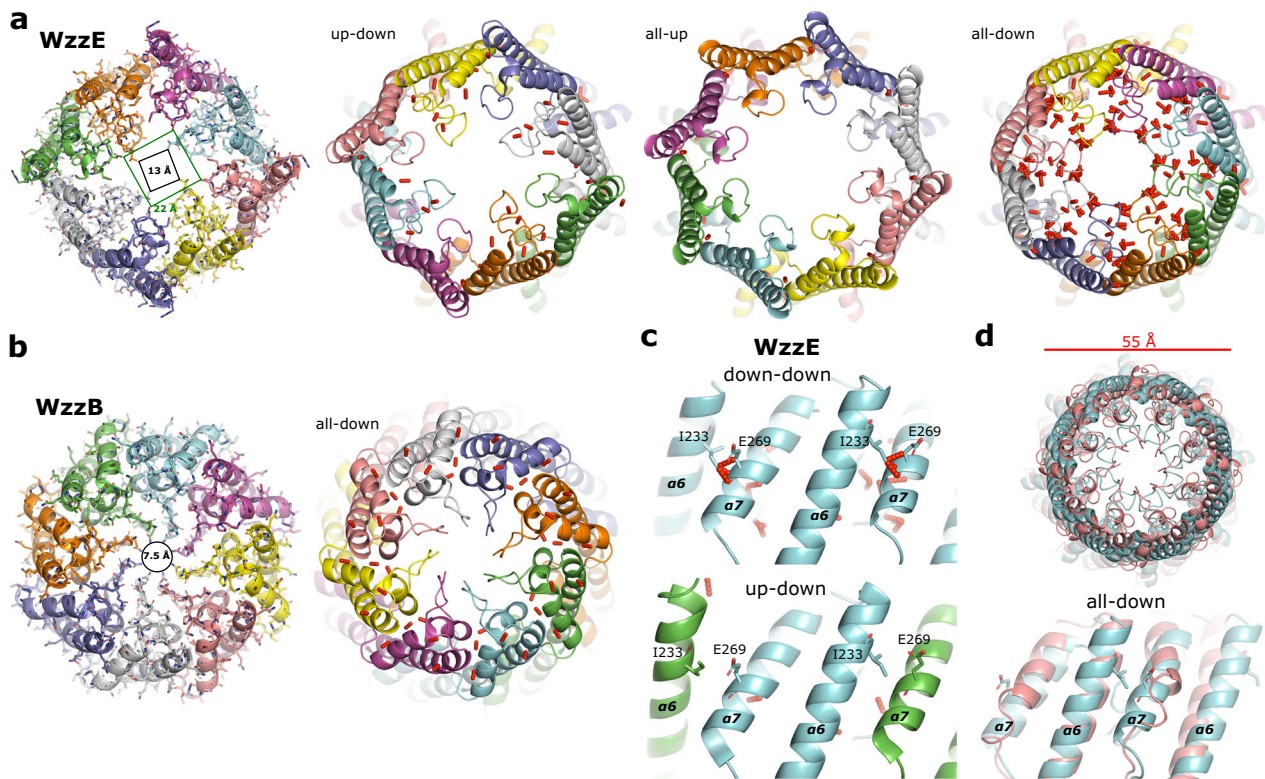

**Fig. 3 Clash comparison of the WzzE and WzzB octamers. a** View looking down from the top of the periplasmic domain of WzzE. The all-up and all-down versions of WzzE were artificially generated by applying C8 symmetry to a single subunit containing an upward and downward facing L4, respectively. **b** View looking down from the top of the periplasmic domain of WzzB (PDB id 6RBG). **c** Zoom of the interaction between α6 and α7 of adjacent subunits of WzzE. Top: the artificial all-down version. Bottom: the naturally occurring alternating up-down version. **d** Overlay of the artificial all-down version of WzzE (cyan) to the naturally occurring all-down WzzB (PDB id 6RBG, salmon). Top: view looking down from the top of the periplasmic domains. Bottom: zoom of the interaction between α6 and α7 of adjacent subunits. In all panels, the red bars represent clashes with a van der Waals distance ratio of less than 0.89. Clashes between the α6 and α7 of adjacent subunits of WzzE will likely prevent two adjacent WzzE downward facing subunits. These clashes are absent in WzzB due to a shortened α7, allowing adjacent downward subunits.

helices can be seen in addition to the simple fluctuation that is typically observed in rigid structures[17]. The analysis reveals a clear bending inward and downward of α6 in upward facing protomers, while the protomers with downward facing L4 loops, α6 appears to be simultaneously bending slightly backward and upwards, albeit much less pronounced compared to the upward versions. Although the buried, more hidden nature of the downward facing L4 loops could be impeding the 3DVA, it could suggest that the downward facing L4 loops maintain their downward position while the upward L4s and flanking α6 and α7 helices are mobile. Additionally, α7 of both protomers can also be seen subtly adjusting to accommodate the movement of the adjacent α6 helices. The direct visualization of movement in the cryo-EM density further verifies the dynamic, alternating loop nature of this region of WzzE. Moreover, the back-and-forth movement of the helices could suggest a ratchet-like mechanism in polysaccharide elongation.

Since the L4 loops are essential for production of long polysaccharides[10,13,14], but are positioned at the top of the periplasmic domain, too far from the inner membrane to physically interact with the polysaccharide polymerase WzyE, it has been suggested that L4 must function as a sugar-binding face for the growing polysaccharide[10]. This structure supports that idea and advances it by suggesting that three adjacent monomers encompassing the down-up-down conformations (Fig. 4a) do in fact create a very attractive potential sugar binding face (Fig. 5a–c). The created binding face is made up of the inner face of α6 (residues 223-239) as well as many of the residues of

the two downwardly oriented L4 loops on opposite sides. The middle L4 loop is accommodated in the upward conformation, and with the exception of E255, is mainly pointing away from the binding face. Together with mainly aliphatic residues this binding face contains an aspartic acid and a number of glutamic acid residues, creating a highly negative pocket (Fig. 5b, c). The alternating nature of the L4 loops create four binding faces per octameric complex, and an electrostatic surface map of the full WzzE (Fig. 5d and Supplementary Fig. 7a) shows an additional highly negatively charged belt around the interior of the periplasmic bell, just above the membrane interface that is separated by 40 Å from the negatively charged binding surface created by the L4 loops (Fig. 4b). In between these two highly negative interior surfaces is a band of flexible loops (Supplementary Fig. 7b). There has been many, seemingly contradictory, studies suggesting that the growing LPS molecule can interact with either the interior or the exterior of the periplasmic bell[10,13,14,18–20]. Although without direct experimental evidence, we cannot rule out the possibility of the ECA chain growing on the outside of WzzE, however, with two internal highly concentrated negative bands and an internal band of flexible, more neutrally charged loops in between, the structure presented herein favors models that place the growing ECA molecule along the interior surface of the Wzz octamer.

**A potential ratchet-like mechanism for polysaccharide elongation.** As mentioned, the alternating arrangement of the L4 loops around the octameric bell would likely create four binding

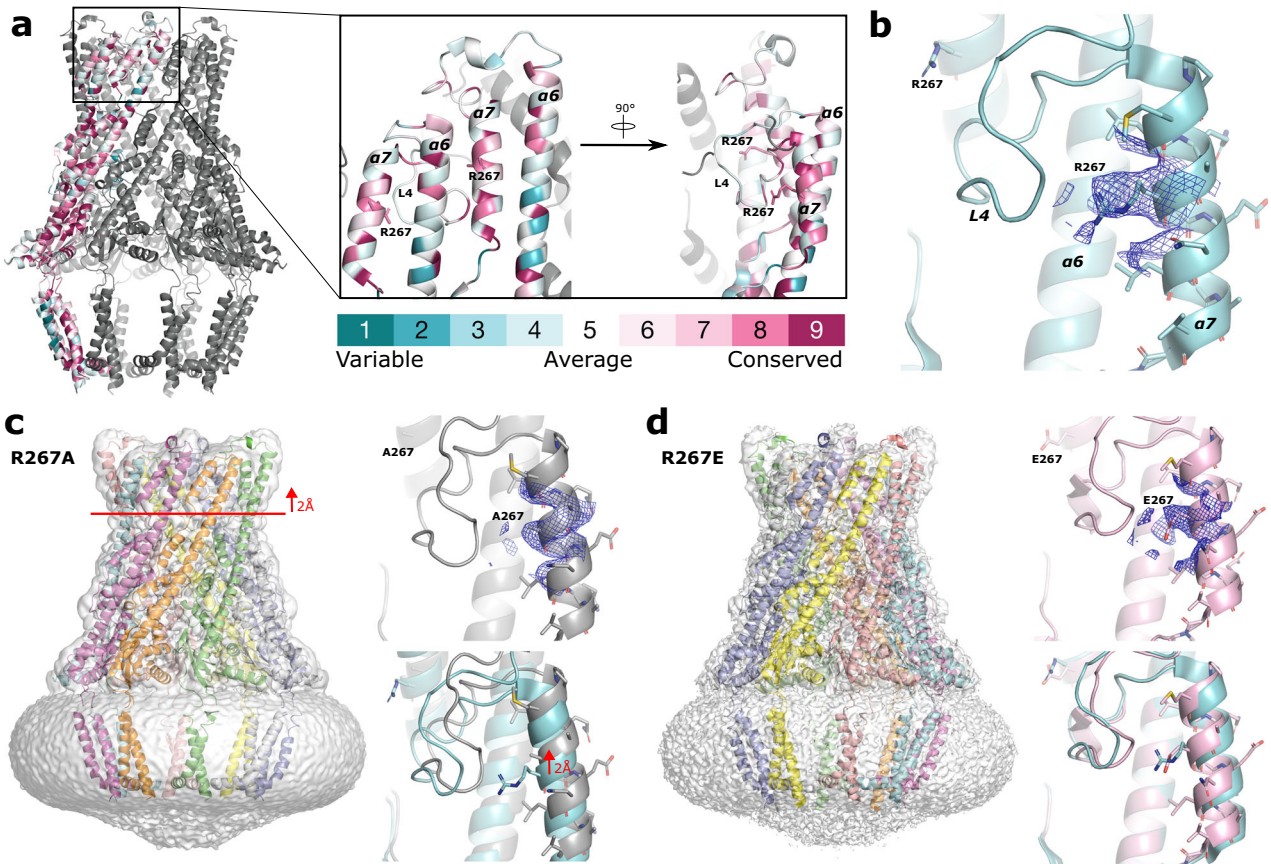

**Fig. 4 Mutations to R267 maintain the alternating L4 arrangement. a** ConSurf analysis of WzzE. Boxed: zoom of the L4 region. R267 is highly conserved. **b** The interaction of R267 with the downward facing L4 with corresponding map density around the region of R267 (blue mesh). **c** Cartoon representation of the full octameric complex of the R267A variant overlaid with the C4 symmetrized map (white) (left). Right: Zoom of the L4 region and corresponding map density around A267 (top) Bottom: overlay with the native WzzE confirms a highly similar structure. R267A displays a small but noticeable 2 Å shift upwards starting around the bottom of α6 (red line). **d** Cartoon representation of the full octameric complex of the R267E variant overlaid with the C4 symmetrized map (white) (left). Right: Zoom of the L4 region and corresponding map density around E267 (top). Bottom: overlay with the native WzzE confirms a highly similar structure.

faces per complex (Fig. 5b). However, if the downward loops are shared between adjacent binding faces a maximum of two growing ECA molecules inside the bell could occupy opposite facing binding faces at any given time since the downward L4 loops would otherwise be engaged with an ECA molecule and thus unavailable to the neighboring binding surface. This could be consistent with reports of the polymerase Wzy forming dimers[21,22], and a recent Alphafold prediction of the polymerase binding on the interior face of Wzz[23]; thus, a polymerase dimer could potentially span the transmembrane interior allowing the elongation of two ECA chains per octamer. This could also suggest that in fact, although this structure is octameric, a minimal down-up-down trimeric WzzE complex would only be required for proper chain length regulation of the ECA molecule. This would be consistent with aspects of the "clock model" that was remarkably hypothesized 30 years ago[24], suggesting that the active form of Wzz is, in fact, a trimer. Furthermore, crosslinking experiments that suggest that Wzz oligomerization is highly dynamic in nature[13,18,25,26] where conflicting reports suggest trimeric, pentameric, hexameric, and octameric arrangements of WzzB and WzzE[7–10,12].

Finally, to further explore the potential of this putative binding face to accommodate sugar molecules, molecular docking was performed using a 2-repeating unit of the terminal end of the negatively charged ECA molecule with a truncated octameric WzzE (residues 201–294) (Supplementary Fig. 8). The resulting

docking trials showed a clear preference for the placement of the sugar into this binding face, with only 2 trials out of 32 being placed outside of this binding face, with remarkably, no sugars being placed on the outside of the bell. Although these docking results need to be confirmed experimentally, they do present an unexpected result of a negatively charged molecule docking to a negatively charged binding face. Though surprising, this could in fact be beneficial for ECA growth as it would create an unstable complex between Wzz and the sugar chain allowing the ECA elongation process to proceed. Any too strong binding would likely result in the elongation process being halted. In fact, if this putative binding face can be confirmed experimentally, it could be exploited with the development of specific positively charged inhibitors that could tightly bind to this negatively charged binding face and essentially block ECA elongation.

The alternating up-down L4 conformations (Supplementary Movie 2) might immediately suggest that ECA polymerization could proceed via a ratchet-type mechanism. This could be somewhat analogous to the unidirectional transport mechanism of LPS molecules from the inner to the outer membrane revealed by recent structural studies on the Lpt system[27,28]. If a rachet-type mechanism[29] is used for LPS transport, it stands to reason it could also be used for LPS biosynthesis. The putative binding faces created by the alternating L4 loops described here could potentially act in a concerted way with the negatively charged ECA molecules allowing them to ratchet upwards (potentially one

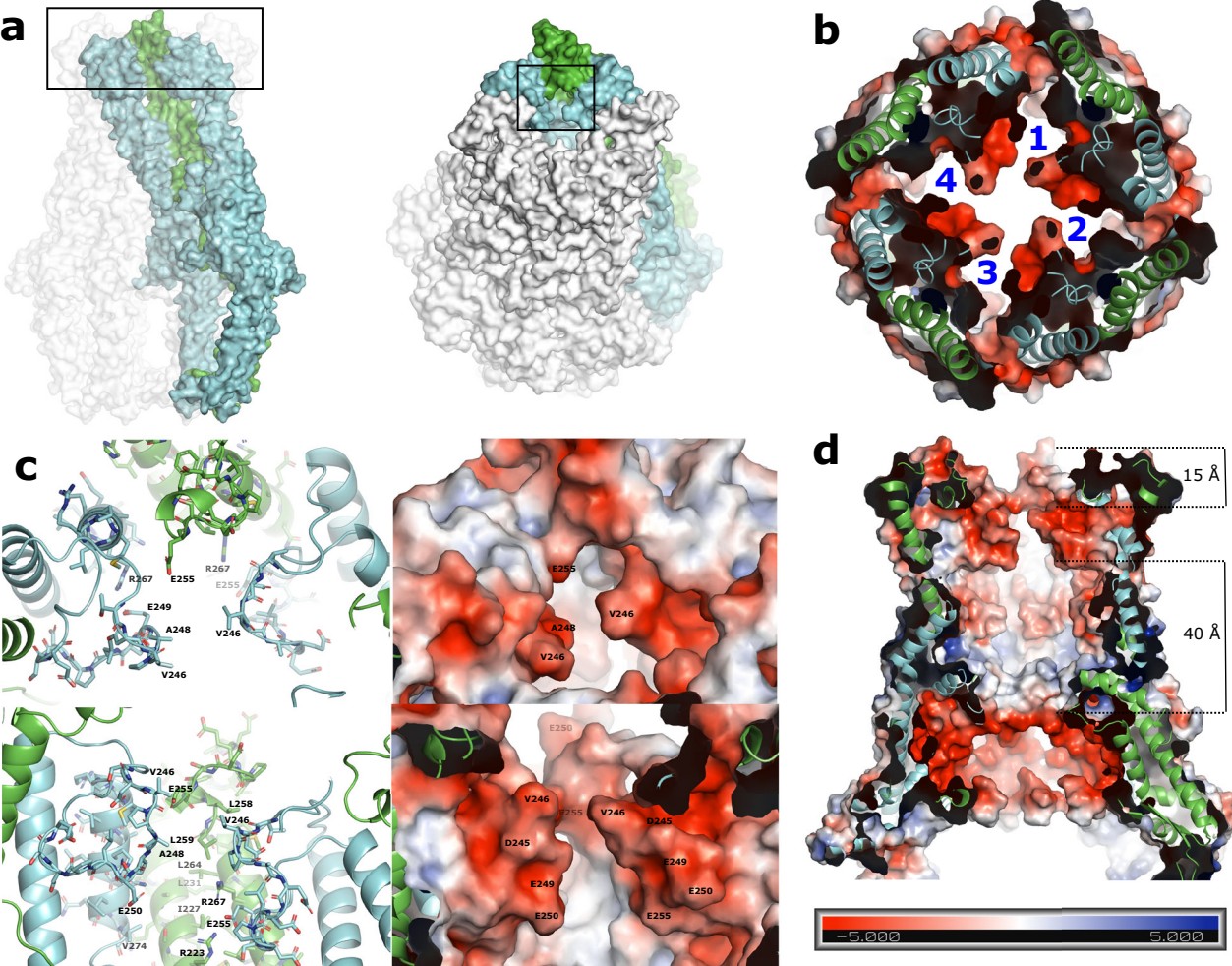

**Fig. 5 Potential binding face created by the up-down nature of the L4 loops. a** Left: Surface representation of the octameric complex. Three colored adjacent protomers make up a potential binding face created by the positioning of the L4 loops, one up (green) surrounded by two down (cyan). The L4 region is boxed. Right: tilted by 45° to display the binding pocket created by the L4 loops (boxed). **b** Electrostatic surface representation showing a highly negative binding face created by the positioning of the L4 loops. View looking down from the top of the periplasmic domain sliced to display the four binding faces created by the positioning of the L4 loops per octameric complex. **c** Zoom of the binding face labeled with its surface residues. Right: electrostatic surface representation showing a highly negative binding face. **d** Electrostatic surface representation of the octameric periplasmic domain sliced to display the interior. In addition to the negative binding face of the L4 loops, the interior surface contains an additional highly negative band just above the membrane interface. A distance of 15 Å separates the top of the upward L4 and the top of the L4 binding face. The two negative bands are separated by 40 Å.

repeating unit at a time) by the dynamic L4 loops. Although the binding face is highly negative in the captured state described here, the movement of L4 would likely alter its surface characteristics and thus its affinity to negatively charged ECA chain potentially permitting a ratcheting type mechanism to proceed. Furthermore, the 3DVA could suggest that the much more stable downward facing L4 loops maintain their downward position to preserve the potential binding face while only the upward L4 ratchets up and down (Fig. 6). This would be consistent with the much higher resolution and lower B-factors of the downward L4 loops compared to the upward facing L4s (Supplementary Figs. 3, 6, and 7b). Additionally, since the clash analysis suggests that adjacent protomers cannot simultaneously be in the downward orientation, the ratcheting of the upward L4 would likely be modulated by repulsion forces between neighboring α6 and α7 helices. Thus, like a ratchet, the upward L4s could only go downward to a point where the repulsive forces from the neighboring α6 and α7 helices would force them upwards again. Finally, any ratcheting-type mechanism of the L4

loops could additionally be aided by the band of flexible, more neutrally charged loops located further down the interior of the periplasmic bell. Although the mechanism of the polysaccharide chain elongation process is currently unknown, the determined structures and analysis presented here provide new insights that will promote new mechanistic ideas to evolve towards elucidating a mechanism of LPS polymerization and length regulation by the Wzy-dependent pathway in general, and the one containing WzzE in particular.

## Methods

**Protein expression and purification**. The full-length WzzE gene was amplified from *E. coli* K12 genomic DNA using standard PCR techniques using the primers EcWzzE_fwd and EcWzzE_rev (Supplementary Table 1) and inserted into a modified pWaldo vector[30] that replaces the GFP reporter with a C-terminal 8×His tag[31] using the restriction sites *XhoI* and *EcoRI* and their sequences confirmed by DNA sequencing. Variants R267A and R267E were created using the appropriate primers (Supplementary Table 1) and the QuikChange Lightening Site-Directed Mutagenesis kit (Agilent) following the manufactures instructions with their sequences confirmed by DNA sequencing. Large-scale protein expression,

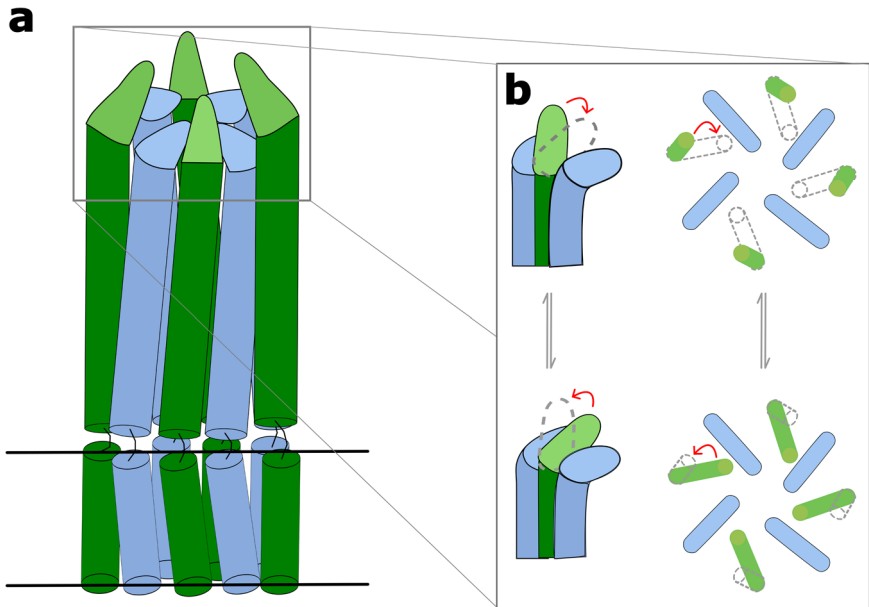

**Fig. 6 Schematic of potential L4 ratcheting in WzzE. a** Schematic model of WzzE with alternating L4 loops. Solid black lines represent the position of the inner membrane. Green: subunits containing the upward oriented L4 loops. Blue: subunits containing the downward oriented L4 loops. **b** Zoom of the L4 loop region. The upward L4s (green) can only go downward to a point where the repulsive forces from the neighboring subunits would force them upwards again. The downward L4s remain relatively stable, maintaining the binding face. Right: view looking down from the top of the periplasmic domain.

solubilization with *n*-Dodecyl-β-D-Maltoside (DDM), and purification was performed as described for EcWzzB[7].

**Gel electrophoresis.** Nu-PAGE Bis–Tris gradient (4–12%) gels (Thermo Fisher Scientific) were loaded with the appropriate samples mixed with loading buffer (40 mM Tris-HCl, pH 6.8, 8% glycerol, 1% SDS, and 1 mg/mL bromophenol blue) and run at 200 V for 40 min in MES running buffer. The gels were stained with a standard 3 mg/mL Coomassie R-250 staining solution and destained with a 10% ethanol, 10% acetic acid solution.

**Cryo-EM sample preparation and data collection.** Purified protein at 3.0 mg/mL (3 μL) was applied to cryo-EM grids (C-flat 2/2-3Au) previously glow-discharged under vacuum for 40 s at 20 mA (PELCO easiGlow), incubated for 30 s, blotted for 3 s, and plunge frozen in liquid ethane using a Vitrobot Mark4 grid freezing device (FEI) with the chamber maintained at 4 °C and 100% relative humidity as previously described[7]. An optimized grid was imaged with a Thermo Scientific Titan Krios G3 electron microscope equipped with either a K2 or K3 camera, operating at 300 kV. Movies of 40 frames each were acquired in electron counting mode with a total exposure does of around 50 electrons/Å² and stacked into a single MRC stack using EPU automatic data collection control software. Detailed data collection parameters can be found in Table S1.

**Cryo-EM image processing.** Motion correction was performed with PatchMotion, contrast transfer function (CTF) parameters were estimated from averaged movies using PatchCTF and initial particle images were selected manually and subjected to 2D classification in cryoSPARC v3[32]. Automatic particle selection was performed with templates from the initial 2D classification. The number of particle images were reduced by further 2D and 3D classifications and heterogeneous refinements (Supplementary Fig. 3). Initial maps were calculated ab initio and the final maps were refined with cryoSPARC's non-uniform refinement feature with and without the application of C4 and C8 symmetry. Detailed image processing parameters can be found in Table 1. Native WzzE maps have been deposited in the Electron Microscopy Data Bank under accession codes EMD-16072 (C1), EMD-16071 (C4), and EMD-16073 (C8), and the mutants R267A and R267E under accession codes EMD-17388 (C1), EMD-17387 (C4), EMD-17390 (C1), EMD-17389 (C4), respectively. 3D variable analysis was performed on native WzzE using particles from a consensus refinement focused around the top of periplasmic bell without the use of symmetry (C1), using a filter resolution of 6 Å and solving for 3 modes. The highest ranked mode of variability is shown in Supplementary Movie 1.

**Model building and refinement.** A final 3.2 Å resolution, C4-symmetrized density map was used for the de novo model building of the asymmetric unit of the WzzE periplasmic domain. Although the WzzE complex is octameric, the asymmetric

unit consists of 2 adjacent WzzE protomers containing an upward and downward state of the L4 loop. Manual building in Coot[33] was performed starting with easily interpretable features from the density map, such as bulky residues and α-helices. After anchor points were established, a full atomic model of the periplasmic domain was then able to be built with the aid of secondary-structure predictions from Jpred[34]. Once completed, the asymmetric model was further improved through iterative rounds of refinement with phenix_real_space_refine[35] and fixing with Coot. After the asymmetric unit was completed in this way, C4 symmetry was applied to the model with the Phenix suite of programs[35]. Due to the low resolution of the transmembrane region, a single copy of an Alphafold2[36] generated model was docked and refined against a C8 symmetrized map. C8 symmetry was then applied to the refined transmembrane domain to generate 8 copies that were then attached to the refined full-octameric model of the periplasmic domain. This full complex containing the transmembrane and periplasmic domains was then refined against the C4 symmetrized map with phenix_real_space_refine, and further improved through iterative rounds of refinement and fixing with Coot. The final model was validated with MolProbity[37] and has been deposited in the Protein Data Bank under the PDB accession code 8BHW. The two R267 variants were built in a similar way, using a single asymmetric unit of native WzzE as a starting model to dock to the respective C4 symmetrized map and improved through iterative rounds of refinement and fixing with Coot. The final models were validated with MolProbity and have been deposited in the Protein Data Bank under the PDB accession codes 8P3O and 8P3P. All figures containing macromolecular structures were made with UCSF Chimera[38] or PyMOL[39]. Detailed model building statistics can be found in Table 1.

**Enterobacterial common antigen modeling and docking.** The terminal region of the ECA polysaccharide containing two repeating units was used for docking. The glycosidic dihedral angles were set to the highest populations from molecular dynamics simulations[40,41] using the set_dihedral command within PyMOL. Prior to docking hydrogen atoms and a net −2 charge was added using the Gasteiger method[42] with the dock prep feature in UCSF Chimera. Docking of the prepared ECA molecule to a truncated octameric WzzE (residues 201–294) was performed with SwissDock[43,44]. Implicit solvent and continuum electrostatics were handled by the Adaptive Poisson-Boltzmann Solver (APBS) and calculated via the PyMOL plugin[45].

**Reporting summary.** Further information on research design is available in the Nature Portfolio Reporting Summary linked to this article.

## Data availability
The cryo-EM density maps and atomic coordinates have been deposited in the Electron Microscopy Data Bank and Protein Data Bank under the accession codes EMD-16071,

EMD-16072, EMD-16073, EMD-17387, EMD-17388, EMD-17389, EMD-17390, and 8BHW, 8P3O, 8P3P respectively. All relevant data supporting the key findings of this study are available within the article and its Supplementary Information files or from the corresponding authors upon reasonable request. Additional maps and source data are available from the corresponding authors upon reasonable request.

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

## Acknowledgements

This research was funded by the Swedish research council (2017-04018, 2017-03703) and the Knut and Alice Wallenberg Foundation (2017.0275 and 2019.0436). Cryo-EM sample screening, optimization, and data collection were performed at the Cryo-EM Swedish National Facility in Stockholm, Sweden funded by the Knut and Alice Wallenberg, Family Erling Persson and Kempe Foundations, SciLifeLab, Stockholm University, and Umeå University.

## Author contributions

B.W. and M.H. conceived and designed the research. B.W. performed gene cloning and protein purification. B.W. performed cryo-EM sample preparation, grid screening, and analysis. B.W. collected the final cryo-EM data, performed image analysis, various calculations with the cryo-EM data, and built the de novo atomic model. B.W. and G.W.

performed ECA modeling and docking. B.W., G.W. and M.H. wrote the manuscript. B.W. prepared data tables, figures and legends.

## Funding

## Competing interests
The authors declare no competing interests.
