## [Peer Review File · Communications Biology]

Reviewers' comments:

Reviewer #1 (Remarks to the Author):

In this study Wiseman et al. present the single-particle cryo-EM structure of the full-length ECA co-polymerase WzzE with its L4 loops resolved in two alternating conformations that are distinct from those observed in the homologous co-polymerase WzzB. The authors use surface electrostatics analysis and docking to suggest that ECA binds to a novel sugar binding site formed by the L4 loops inside the periplasmic bell of WzzE. The authors propose that the ECA polymer grows along the interior of the WzzE periplasmic bell and that the alternating L4 loops could act as a ratchet promoting unidirectional transport of the ECA chain.

The manuscript is concise and well-written, and the data are presented in a logical manner. The cryo-EM data collection, processing and model building is described with sufficient detail. The authors provide a data processing workflow as well as global and local resolution estimates for the maps in line with current best practice in single-particle cryo-EM. The validation files do not reveal any major issues with the model or the cryo-EM maps. The structure will be of interest to those working on the mechanistic aspects of LPS biosynthesis as the L4 loops are required for production of appropriate length LPS. The structure of WzzE with the resolved L4 loops does not offer direct insight into their role in regulating LPS length, but it will inspire future functional work. The authors try to speculate what role the alternating L4 loops might play in the co-polymerase function, but do not show anything convincing.

The major issues I have are to do with the docking and lack of context.

1. In L195-6 the authors state that "a net -2 charge" was applied to the ECA model. Do the authors not find it surprising that the negatively charged ECA molecule is docked into a highly negatively charged pocket (as shown in Figure 4c)? This raises doubts regarding the usefulness of the docking results, and if there is an explanation for these results, the authors do not provide it. Would an oligosaccharide that is completely unrelated to ECA be docked to the same pocket (e.g. by virtue of satisfying a hydrogen bond scoring function in the docking algorithm)?
2. The docking that the authors perform is not truly 'blind' because they supply only part of the protein model (residues 201-294). What are the docking results using a full-length WzzE model? Or, if computationally too demanding, at least using three full-length protomers, which the authors propose is the minimum ECA-binding unit?
3. The authors seem to be committed to the idea that the ECA chain grows inside WzzE without much evidence, and contrary to the other proposed models where the ECA chain binds on the outside of the periplasmic bell (e.g. PMID: 28434914). The authors should comment on whether their structure is compatible with alternative models.
4. I am not familiar with the literature as I am not an expert on the Wzy-dependent biosynthesis pathway. Have there been any WzzE L4 variants studied that could illuminate how specific mutations might interfere with the proposed "alternating loop" model?

Minor:

5. Please indicate in the text which residues of WzzE form L4.
6. It is not clear why L4 has to alternate between adjacent protomers. It would be informative to have a figure with all-down and all-up octamers that would clearly show how these arrangements would lead to clashes between adjacent protomers.
7. L95: please change "newly discovered" to "putative" or similar.
8. L103: there are clearly separated positively and negatively charged patches on the surface as shown in Supplementary Figure 4b, so I think it's misleading to say that the outer surface is "relatively neutrally charged."
9. L118-120: did the authors observe any alternative oligomeric states in their cryo-EM data analysis (apart from non-physiological dimers of octamers)? If a single oligomeric state was observed, this

would be inconsistent with dynamic oligomerization of WzzE based on crosslinking, even if a down-up-down trimer is the minimal ECA chain regulating unit.

10. L127-129: please explain how negatively charged faces would facilitate ratcheting of negatively charged ECA molecules.
11. Figure 2: the WzzB molecule is very hard to see. Also, showing unlabelled stick models for all side chains makes panels b and c hard to interpret. Maybe just show the V246 side chain?
12. Figure 3: please state the PDB ID of the WzzB structure in the legend. Also, it seems like a wasted opportunity to not show a sequence alignment of the L4 loop of WzzE from different organisms. Are some residues more conserved than others?
13. Figure 4: it would be useful if the authors indicated in panel a which part of the full-length model corresponds to residues 201-294 used in docking.
14. Supplementary Figure 4: panel a is incomprehensible. Maybe show the top scoring docking results as separate panels?
15. Supplementary Video 1: the authors show morphing of the protomers of the octameric complex between the L4 up and down conformations. But is there any evidence that protomers alternate between the two states, apart from poorly resolved L4 in previous structures? Have the authors tried 3D variability analysis in cryoSPARC? I suspect that the L4 might be too small to get meaningful results, but it would be more convincing to see dynamics in the cryo-EM density, even if at low resolution.

Reviewer #2 (Remarks to the Author):

The manuscript by Wiseman et al. determined a cryo-EM structure of the full-length octameric polysaccharide co-polymerase WzzE at a resolution of 3.2Å. With the un-symmetry (C1) or C4 symmetry application, the authors revealed an alternating up and down conformations of the L4 loops of WzzE, which was not reported before for WzzE and the homolog WzzB. The authors proposed a potential sugar binding surface at a trimer of L4 loops encompassing the down-up-down conformation by showing the negatively charged pocket created by the two downwardly oriented L4 loops and the inner face alpha 6, and performing blind docking to show the placement of sugar on the proposed binding surface instead of the outside of the complex. Based on the alternating up and down conformational arrangement and binding potential of the L4 loops, the authors propose a trimeric active unit of WzzE and a ratchet-type mechanism for unidirectional ECA polymerization.

The work by Wiseman et al. revealed a novel conformational change in the flexible periplasmic loop of this type of polymerase, which is very interesting and leads to new discussion for the mechanism of LPS biosynthesis and transport. Based on the results and discussions I have few questions:

1. The alternating up and down conformations in L4 loops reported here was not seen in WzzB structure. Do the authors think WzzB acts through a mechanism that is distinct from WzzE? or the absence of the feature was due to different methods of cryo-EM data process?
2. Un-symmetry C1 or symmetry C4 and C8 were used in cryo-electron microscopy data processing. Comparing to C1 and C4 maps, the L4 loops were not resolved in the map processed with C8, but the overall resolution is higher. Are there any differences in other resolved parts comparing the C8 with C1 or C4 structures? C1 applied map revealed the L4 Loops, why the authors did not used this map to build the model? Is there any difference between the models processed by C1 and C4? It would be clear if the authors can superimpose these structures in supplementary to clarify.
3. The author suggests that L4 loop and surrounding negatively charged amino acids participate in polysaccharide binding and regulation. Is there any in vitro binding test and mutagenic assay can be done to prove that these amino acids are specifically important for polysaccharide binding?

4. The authors claimed that the alternating conformational changes of the L4 loops were due to high flexibility. Does this mean that the upwardly oriented L4 loop is randomly positioned? What decides that the middle loop of the trimer is upwardly oriented? Is it the size of the substrate sugar? Is it possible to have the down-up-up-down tetrameric L4 active unit when the binding sugar unit is larger?

We thank the reviewers and editor for their time, constructive comments and suggestions to substantially improve this manuscript. Below we detail our replies and revisions to address the reviewers' comments.

Reviewer #1 (Remarks to the Author):

In this study Wiseman et al. present the single-particle cryo-EM structure of the full-length ECA co-polymerase WzzE with its L4 loops resolved in two alternating conformations that are distinct from those observed in the homologous co-polymerase WzzB. The authors use surface electrostatics analysis and docking to suggest that ECA binds to a novel sugar binding site formed by the L4 loops inside the periplasmic bell of WzzE. The authors propose that the ECA polymer grows along the interior of the WzzE periplasmic bell and that the alternating L4 loops could act as a ratchet promoting unidirectional transport of the ECA chain.

The manuscript is concise and well-written, and the data are presented in a logical manner. The cryo-EM data collection, processing and model building is described with sufficient detail. The authors provide a data processing workflow as well as global and local resolution estimates for the maps in line with current best practice in single-particle cryo-EM. The validation files do not reveal any major issues with the model or the cryo-EM maps. The structure will be of interest to those working on the mechanistic aspects of LPS biosynthesis as the L4 loops are required for production of appropriate length LPS. The structure of WzzE with the resolved L4 loops does not offer direct insight into their role in regulating LPS length, but it will inspire future functional work. The authors try to speculate what role the alternating L4 loops might play in the co-polymerase function, but do not show anything convincing.

The major issues I have are to do with the docking and lack of context.

1. In L195-6 the authors state that "a net -2 charge" was applied to the ECA model. Do the authors not find it surprising that the negatively charged ECA molecule is docked into a highly negatively charged pocket (as shown in Figure 4c)? This raises doubts regarding the usefulness of the docking results, and if there is an explanation for these results, the authors do not provide it. Would an oligosaccharide that is completely unrelated to ECA be docked to the same pocket (e.g. by virtue of satisfying a hydrogen bond scoring function in the docking algorithm)? **Yes, we agree that any docking results are speculative and need to be confirmed or rejected with future experiments. We have revised this section, more clearly stating that these are speculations to be verified or rejected in future experiments. Also, we have moved the docking results to the supplementary information; with the improved manuscript and additional data and analysis we have placed less importance on these docking results. However, we have decided to still include them, as we do feel the docking results are, if nothing else, interesting and could be used to formulate ideas and hypotheses for future experiments. We have included in the manuscript a brief discussion as to why we believe a negatively**

charged ECA docking to a negative binding pocket would be beneficial for any LPS elongation mechanism.

2. The docking that the authors perform is not truly 'blind' because they supply only part of the protein model (residues 201-294). What are the docking results using a full-length WzzE model? Or, if computationally too demanding, at least using three full-length protomers, which the authors propose is the minimum ECA-binding unit?

We agree. We removed the word "blind". Indeed, docking to the full length WzzE was too computationally demanding. Even using only three full length protomers was also too computationally demanding.

3. The authors seem to be committed to the idea that the ECA chain grows inside WzzE without much evidence, and contrary to the other proposed models where the ECA chain binds on the outside of the periplasmic bell (e.g. PMID: 28434914). The authors should comment on whether their structure is compatible with alternative models. Yes, this is of course still possible, there are many seemingly contradictory, studies regarding the location of the growing LPS molecule (referenced in the manuscript). We have further commented on the possibility of ECA chains along the outside of WzzE. Our current data suggests the ECA chain interacts on the inside and adds to this ongoing discussion. Like any previously proposed models, the current suggestion should also be subject to future studies.

4. I am not familiar with the literature as I am not an expert on the Wzy-dependent biosynthesis pathway. Have there been any WzzE L4 variants studied that could illuminate how specific mutations might interfere with the proposed "alternating loop" model? To our knowledge there are no mutational studies to date on WzzE that would interfere with the alternating loops, however a mutational study on WzzB that we refer to in the revised manuscript might be consistent with our findings (PMID: 2897668). Also, in the additional experiments presented in the revised paper, we specifically tried to interfere with the alternating loop by mutating the highly conserved R267 that interacts with the L4 loop. However, both variants still displayed the alternating arrangement supporting that this is a robust structural feature.

Minor:

5. Please indicate in the text which residues of WzzE form L4. Ok. See L74.

6. It is not clear why L4 has to alternate between adjacent protomers. It would be informative to have a figure with all-down and all-up octamers that would clearly show how these arrangements would lead to clashes between adjacent protomers. Thanks, this was very useful! We have done this analysis and provided a figure (Fig. 3) and discussion.

7. L95: please change "newly discovered" to "putative" or similar. Ok.

8. L103: there are clearly separated positively and negatively charged patches on the surface as shown in Supplementary Figure 4b, so I think it's misleading to say that the outer surface is "relatively neutrally charged." Yes, that is true. We have removed that phrase.

9. L118-120: did the authors observe any alternative oligomeric states in their cryo-EM

data analysis (apart from non-physiological dimers of octamers)? If a single oligomeric state was observed, this would be inconsistent with dynamic oligomerization of WzzE based on crosslinking, even if a down-up-down trimer is the minimal ECA chain regulating unit. **We did not observe anything other than octamers and the dimer of octamers during our cryo-em single particle analysis. But of course, that does not mean that they do not exist. As described for WzzB, this sample also likely contains multiple unstable oligomeric states that could not be accurately visualized using the single-particle technique. As the reviewer is surely aware, in structural biology the most stable state is the one most often studied. In this case, similar to WzzB the octamer is likely the most stable state and thus the one that we are able to visualize in our micrographs.**

10. L127-129: please explain how negatively charged faces would facilitate ratcheting of negatively charged ECA molecules. **We have tried to further explain this. Please see discussion. Briefly, the movement of L4 would alter the surface characteristics of the binding face as the loops morph between the 2 states.**

11. Figure 2: the WzzB molecule is very hard to see. **We have increased the contrast of WzzB, we hope it is more visible now.** Also, showing unlabelled stick models for all side chains makes panels b and c hard to interpret. Maybe just show the V246 side chain? **We agree it is unclear. We have shown only the V246 side chain now.**

12. Figure 3: please state the PDB ID of the WzzB structure in the legend. **Ok.** Also, it seems like a wasted opportunity to not show a sequence alignment of the L4 loop of WzzE from different organisms. Are some residues more conserved than others? **Ok. We have performed a ConSurf analysis. See Fig. 4 and Fig S5 for details. This turned out to also be very useful. Thanks!**

13. Figure 4: it would be useful if the authors indicated in panel a which part of the full-length model corresponds to residues 201-294 used in docking. **Ok. See supplementary fig. 8.**

14. Supplementary Figure 4: panel a is incomprehensible. Maybe show the top scoring docking results as separate panels? **Ok. We have included a few examples of the docking results. See supplementary fig. 8.**

15. Supplementary Video 1: the authors show morphing of the protomers of the octameric complex between the L4 up and down conformations. But is there any evidence that protomers alternate between the two states, apart from poorly resolved L4 in previous structures? Have the authors tried 3D variability analysis in cryoSPARC? I suspect that the L4 might be too small to get meaningful results, but it would be more convincing to see dynamics in the cryo-EM density, even if at low resolution. **We have performed 3D variable analysis. Indeed, as you point out, despite the relatively small size of the L4 loops, this analysis suggest that the L4 loops do appear to be much more dynamic than the exterior of the bell and the upward facing L4 does appear to move. We have included a brief discussion and movie of the analysis: Supplementary video 1. Based on the 3D variability and clash analysis we also discuss the possibility of an alternative ratcheting mechanism in the text and the new figure 6.**

Reviewer #2 (Remarks to the Author):

The manuscript by Wiseman et al. determined a cryo-EM structure of the full-length octameric polysaccharide co-polymerase WzzE at a resolution of 3.2Å. With the un-symmetry (C1) or C4 symmetry application, the authors revealed an alternating up and down conformations of the L4 loops of WzzE, which was not reported before for WzzE and the homolog WzzB. The authors proposed a potential sugar binding surface at a trimer of L4 loops encompassing the down-up-down conformation by showing the negatively charged pocket created by the two downwardly oriented L4 loops and the inner face alpha 6, and performing blind docking to show the placement of sugar on the proposed binding surface instead of the outside of the complex. Based on the alternating up and down conformational arrangement and binding potential of the L4 loops, the authors propose a trimeric active unit of WzzE and a ratchet-type mechanism for unidirectional ECA polymerization.

The work by Wiseman et al. revealed a novel conformational change in the flexible periplasmic loop of this type of polymerase, which is very interesting and leads to new discussion for the mechanism of LPS biosynthesis and transport. Based on the results and discussions I have few questions:

1. The alternating up and down conformations in L4 loops reported here was not seen in WzzB structure. **The WzzE and WzzB structures were processed similarly. Starting maps were calculated *ab initio* and reference-free without any user interference. Final maps have been validated using standard current cryo-em practices with half-maps and by the gold-standard Fourier shell correlation. Thus, the novel features seen in this structure are absolutely not an artifact of data processing. Additionally, the two R267 variant structures presented in the revised paper were solved independently and arrived at the same structure as native WzzE, further supporting an alternating arrangement.**

Do the authors think WzzB acts through a mechanism that is distinct from WzzE? or the absence of the feature was due to different methods of cryo-EM data process? **It is difficult to speculate whether WzzB and WzzE have distinct mechanisms. We think fundamentally the mechanisms are the same, for example the interaction between the polymerase and co-polymerase and the LPS elongation mechanism are likely conserved. Fine details like how the ECA/LPS interacts with the L4 loops might differ between the two. But without additional information it is difficult to speculate at this time.**

2. Un-symmetry C1 or symmetry C4 and C8 were used in cryo-electron microscopy data processing. Comparing to C1 and C4 maps, the L4 loops were not resolved in the map processed with C8, but the overall resolution is higher. Are there any differences in other resolved parts comparing the C8 with C1 or C4 structures? C1 applied map revealed the L4 Loops, why the authors did not used this map to build the model? Is there any difference between the models processed by C1 and C4? It would be clear if the authors can superimpose these structures in supplementary to clarify. **We did not see any significant differences between the three symmetries (other than the averaging out of the L4 loops in the C8 maps as mentioned in the text). We used the C4 map to simplify**

model building as described in the M&M section. To address your valid point, we refined the model against the C1 map, and manually checked the entire structure. In short, we do not see any major differences between the subunits of the C1 refined structure (other than the L4 loops of course). In slightly lower resolution areas such as the L4 loops and other interior loops, side-chain placement is ambiguous making any potential differences difficult to detect. Additionally, the highly symmetrical nature of the complex likely results in the miss-alignment of particles along the C8/C4 axis making this analysis somewhat difficult in our opinion.

3. The author suggests that L4 loop and surrounding negatively charged amino acids participate in polysaccharide binding and regulation. Is there any *in vitro* binding test and mutagenic assay can be done to prove that these amino acids are specifically important for polysaccharide binding? Unfortunately to our knowledge there is currently no *in vitro* binding test. Hopefully this can be developed in the future to complement our structural studies.

4. The authors claimed that the alternating conformational changes of the L4 loops were due to high flexibility. Does this mean that the upwardly oriented L4 loop is randomly positioned? What decides that the middle loop of the trimer is upwardly oriented? Is it the size of the substrate sugar? Is it possible to have the down-up-up-down tetrameric L4 active unit when the binding sugar unit is larger? We do not see any evidence of anything other than an alternating up-down arrangement of the L4 loops. We have performed multiple rounds of 3D classification (on the full WzzE structure as well as focused around the L4 loops) and have not seen any evidence of other arrangements, i.e., no up-down-down-up, up-up-down-down, down-up-up-down, etc. Additionally, based on our clash analysis (Fig. 3) we believe that adjacent downward subunits to be unlikely. Similarly based on our clash analysis it should be possible to have adjacent upward subunits, but we do not see any experimental evidence of that in our single-particle data. We speculate as to the reason why in the revised manuscript.

REVIEWERS' COMMENTS:

Reviewer #1 (Remarks to the Author):

In the revised version of the manuscript "Alternating L4 loop architecture of the bacterial polysaccharide co-polymerase WzzE," Wiseman et al. include single particle cryo-EM structures of two WzzE R267 variants, protomer clash and ConSurf analyses, as well as 3D variability analysis of the L4 region. The variants did not have any effect on L4 mobility, but the bioinformatics analyses highlight interesting avenues for further study. The 3D variability analysis supports the authors' conclusion that the L4 loops, at least the upward-facing ones, are mobile. The speculation about the ECA binding site is now clearly delineated from the experimental data.

All of my previously made points have been addressed by the authors. The revised manuscript should be published in Communications Biology.

Reviewer #2 (Remarks to the Author):

The manuscript has been significantly improved and the authors have fully addressed my questions. I have no further questions, thus support the publication of the manuscript.